# Ingredients and Process Affect the Structural Quality of Recombinant Plant-Based Meat Alternatives and Their Components

**DOI:** 10.3390/foods11152202

**Published:** 2022-07-25

**Authors:** Di Zhao, Lu Huang, He Li, Yuqing Ren, Jinnuo Cao, Tianyu Zhang, Xinqi Liu

**Affiliations:** 1National Soybean Processing Industry Technology Innovation Center, Beijing Technology and Business University (BTBU), Beijing 100048, China; zhaodi22121@163.com (D.Z.); huanglulu1119@163.com (L.H.); ryq512@163.com (Y.R.); liuxinqi@btbu.edu.cn (X.L.); 2Plant Meat (Hangzhou) Health Technology Limited Company, Hangzhou 311121, China; jinnuocao@163.com; 3Shandong Gulin Food Technology Limited Company, Yantai 264010, China; zhangtiannyu@163.com

**Keywords:** recombinant plant-based meat alternatives, tissue protein, simulated fat, structural quality, assembly molding

## Abstract

Recombinant plant-based meat alternatives are a kind of product that simulates animal meat with complete structure by assembling plant-tissue protein and other plant-based ingredients. The market is growing rapidly and appears to have a promising future due to the broad culinary applicability of such products. Based on the analysis and summary of the relevant literature in the recent five years, this review summarizes the effects of raw materials and production methods on the structure and quality of specific components (tissue protein and simulated fat) in plant-based meat alternatives. Furthermore, the important roles of tissue and simulated fat as the main components of recombinant plant-based meat alternatives are further elucidated herein. In this paper, the factors affecting the structure and quality of plant-based meat alternatives are analyzed from part to whole, with the aim of contributing to the structural optimization and providing reference for the future development of the plant meat industry.

## 1. Introduction

As the global population continues to grow and natural resources are progressively reduced, the demand for protein supplies is increasingly strained [1,2]. Traditional livestock farming is gradually becoming overwhelmed as the primary source of meat protein, with insufficient livestock land resources, environmental pollution, the greenhouse effect, veterinary drugs, hormone residues and animal welfare concerns among those posing limiting factors for the industry [3,4]. Furthermore, the availability of meat protein (an average of 1 kg of meat protein for every 6 kg of plant protein) [5] is not sufficient to fulfill the demand of the world’s steadily growing population. Consequently, recently developed plant-protein-based meat substitutes are increasingly filling the protein supply gap, increasing the proportion of plant protein going directly to the table.

Plant-based meat alternatives, a type of artificial meat, are composed of plant protein processed either with or without the addition of auxiliary materials. This class of meat substitutes is considered similar in texture, flavor, morphology and other characteristics of animal meat products. In comparison to some of the other currently available meat substitutes, plant-based meat alternatives have gained a certain consumer base and are easily accepted by the public. Moreover, the production and processing of plant-based meat alternatives exert low impact on the environment, which is consistent with the concept of sustainable development and contributes to the realization of ‘carbon neutrality’ in the food industry [6]. The market for plant-based meat alternatives has exceeded US $5 billion and is expected to reach US $6.4 billion by early 2023, with more than 400 million consumers worldwide enjoying plant-based meat alternative products [7]. Thus, the plant-based meat alternatives market is burgeoning and has great potential for further development. One of the most important factors influencing the consumer acceptability of plant-based meat alternatives is its structural similarity to animal meat products. The reproduction of the hierarchical structure of known meat tissues helps to simulate their functional and sensory properties [8]. Therefore, overcoming the technological obstacles to mimicking the structure of meat products to meet the needs and expectations of customers is a key direction in the development of plant-based meat alternatives [9].

Recombinant plant-based meat alternatives are produced by adding blocks of simulated fat or simulated fat binder to the basis of tissue protein to simulate structurally intact animal meat [10]. Compared with the structure of single-tissue protein, the structure of recombinant plant-based meat alternatives is more complete and more suitable for cooking, successfully simulating the appearance, texture and flavor of animal meat products, and has, consequently, become a hot spot for development in the food industry. At present, recombinant plant-based meat alternative products include relatively simple mincemeat products, such as plant-based burger patties, plant-based sausages and plant-based meatballs, while complete meat blocks, such as pork pancetta and snowy beef, are neither widely simulated nor studied due to the complexity of their compositional structure [11]. Similar products did appear at the early stage of recombinant plant protein development, due to the demand for such meat products from vegetarian; however, these were mostly distorted and poorly simulated. In recent years, based on new developments in tissue-protein-related technologies, research related to the blocking of simulated fat and binders has increased progressively, and whole recombinant plant-based meat alternative products have gradually emerged [12].

The factors affecting the structural quality of plant-based meat alternatives are complex and diverse. Previous studies and related reviews have focused on the general characteristics of plant-based meat alternatives in a broad sense, so there is currently a lack of analytical data regarding specific types of plant-based meat alternatives. [13,14] Additionally, most studies have explored the effects of the raw material components on plant-based meat alternatives in general, rather than the effects of raw material on specific components (such as tissue protein or simulated fat) or the effects of individual components on the overall characteristics of plant-based meat alternatives. Therefore, this review emphasizes the effects of different raw material components as well as various production methods on tissue protein, simulated fat and recombinant plant-based meat alternatives as a whole from the perspective of structural optimization of plant-based meat alternatives (as shown in Figure 1), and with the aim of aiding the future development and structural quality optimization of recombinant plant-based meat alternatives on an industrial scale.

## 2. Factors Affecting the Structural Quality of Tissue Protein

Tissue proteins or textured vegetable proteins (with a lean fiber-like structure and chewiness) form the main skeletal structure of plant-based meat alternative products. These can be obtained through techniques such as extrusion, shearing, spinning, freezing structuring and three-dimensional (3D) printing [18]. The main constituents of tissue proteins are proteins, lipids, water, carbohydrates, flavoring agents and coloring agents. The choice of plant protein raw materials, with their various processing characteristics, prior to processing is a decisive factor in structural quality differentiation. During processing, the appropriate lipid and water content are essential to ensuring good structural and sensory properties. Finally, the structure of the simulated meat forms during the process, and different forming methods exert various influences on the final structure of the tissue protein. Thus, this section focuses on the collation of the main components (proteins, lipids and water) and the effect of different production methods on the structural quality of tissue proteins in recombinant plant-based meat alternatives.

### 2.1. Effects of Protein on the Structural Quality of Tissue Protein

Protein is the most important factor impacting the structural quality of tissue protein, as it is the main component of the fibrous structure that mimics meat products. Protein melting and denaturation are prerequisites for the formation of histochemistry. During processing, the combined actions of temperature, shear, pressure and water melt the plant proteins completely, with the molecular chains unfolding in the flow direction, exposing the sulfhydryl groups and creating new disulfide bonds through oxidation. In this way, the protein denatures, losing its original structure, and depolymerizes from the natural spherical aggregated state to form an anisotropic structure. This molten state includes the action of molecules with other substances, in which the protein molecules are rearranged. After sufficient homogenization and oriented arrangement, the mixture is cooled down and characterized, with the protein having been converted into a tissue protein with a structure similar to that of animal meat fibers [19]. Thus, to facilitate this process, it is essential that the proteins used as raw materials contain sufficient sulfur-containing amino acids in their globulin molecular chains [20]. In the next section, the differences and advantages of various raw protein materials, as well as their impact on the structural quality of tissue protein, will be collated and compared.

Table 1, below, presents a comparison of the nutritional properties, processing characteristics and respective limiting factors in the production of different types of raw protein. The practical applications of these products are also summarized to provide a foundation of reference for the selection of raw protein materials for recombinant plant-based meat alternatives.

#### 2.1.1. Legume Protein

Legumes are currently the most important source of protein in the production and processing of tissue protein. Soybeans and other legumes differ in their main components, nutritional values and processing characteristics. For example, soybean is extremely low in carbohydrates, high in fat and protein, and has a more balanced amino acid ratio. Protein content in pea is slightly lower than soybean, but it is one of the major plant protein resources due to the high yield. This section summarizes and analyzes the potential of legume proteins to be used in the production of tissue proteins, using two different categories of legumes as examples.

Soybean protein is a common, high-quality plant protein that can readily replace animal protein [37]. The origin of tofu, which is made from coagulated soy milk, can be traced all the way back to A.D. 965 [38]. Due to the wide availability of high-quality raw soybean materials, the relatively established extraction method and the excellent processing characteristics of anisotropic fiber structures, they are currently widely utilized in the production of tissue protein for recombinant plant-based meat alternatives. The globulin molecular chain in soybean protein contains abundant sulfur-containing amino acids for the manufacturing of tissue protein, which tend to open the chain when subjected to thermal shear, thereby exposing molecular binding sites and contributing to further oxidation to form disulfide bonds and promote the formation of fibrous meshwork. The main forces at work in this organizing process are disulfide bonds, hydrogen bonds, hydrophobic forces and electrostatic interactions, with various proportions of each bond resulting in varied levels of thermal reversibility and structural stiffness [39]. The complex changes in production involve changes in both covalent and noncovalent interactions, with disulfide bonds playing a more important role in the formation of rigid structures and fiber textures than noncovalent interactions [40].

Among legume proteins, pea protein is another of the main raw materials occupying an important position in the plant-based meat alternatives market. This is due to its characteristics of being a non-genetically modified organism (GMO), non-allergenic and estrogen free, with high bioavailability, local cultivation, high yield and low cost [20]. Peas also have considerable development potential due to their processing characteristics and functional food advantages, yet their commercial utilization is currently limited, mainly due to their less-ideal organoleptic properties. However, the processing properties of pea protein can be effectively improved through protein modification. According to a study for the sequential enzyme modification of pea protein, the covalent linking of the protein to hydrophilic polysaccharides can significantly improve the solubility, emulsification properties, water and oil holding capacity of pea protein to form a strong, fibrous and tough tissue protein [41,42]. In another study by Fang et al. [43], pea proteins treated with glutaminase showed higher flexibility, homogeneity and dispersion, and reduced the bean odor and lumpy flocculation in tissue proteins compared to untreated pea proteins. Sandoval et al. [44], who thoroughly investigated the interactions of pea proteins and the mechanism of fibrous structure formation during their processing, concluded that the protein macromolecular network was formed mainly through disulfide bonds during extrusion and cooling, and the formation of the fibrous structure was further facilitated by spinodal phase separation. Thus, protein modification offers a promising approach to enhance the suitability of pea-protein processing, and the obtained modified pea protein has excellent potential in protein tissue processing.

#### 2.1.2. Grain Protein

The protein content of grains is lower than that of legumes, with poor digestibility and unsatisfactory amino acid composition, particularly in the lack of lysine and threonine [45]. Indeed, the intake of a single category of protein often does not meet the full nutritional needs of the body. For example, most legume proteins are low in methionine and cysteine, while grain proteins are often low in lysine. Consequently, grain proteins require supplementation to improve their biological value in plant-based meat alternatives and to balance the ratio of amino acids in the final product.

Following hydration, wheat protein forms a 3D protein network structure maintained by disulfide bonds, which plays an important role in simulating the fibrous structure of meat, increasing the viscoelasticity and hardness of the product, and improving its color stability, juiciness and water retention [28]. In one study, for example, the extrudate obtained by adding 30% wheat gluten protein under high moisture extrusion conditions was found to have improved organization, firmness and chewiness [46]. The texture was closer to that of real meat, and the viscoelasticity, formability and film-forming properties were enhanced. Guo et al. [47] demonstrated that the proportion of α-helices and reverse parallel β-folds (associated with hydrogen bonding and representing stable secondary structures of proteins) decreased, while the proportion of loose β-turned and randomly curled structures increased in tissue, resulting in improved intramolecular hydrogen-bonding interactions, while promoting water and the retention of flavor substances. Wheat protein plays an important role in increasing disulfide bonding and promoting the formation of fibrous structures because hydrogen bonding is the main interaction force between forming and stabilizing proteins, and disulfide bonding is the key force in forming the fibrous structure of high-moisture extruded tissue proteins [46]. Consequently, wheat protein (with its good processing characteristics) is a more commonly used grain protein, which can be added to plant-based meat alternatives as a second composite protein raw material with a complementary amino acid balance, and has an optimal effect on its histochemical structure quality, which is conducive to the development of high-quality plant-based meat alternative products.

#### 2.1.3. Non-Plant Classified Protein

Although fungi are not essentially plants, meat analogs made from fungal proteins are often defined as plant-based meat alternatives. Fusarium, Agaricus bisporus and other fungal proteins are sources of high-quality plant-based meat alternatives [48]. Mycelium has a fibrous-like microstructure, and the fungal protein developed from mycelium is similar to that of meat protein, with excellent hardness, elasticity, chewiness and outstanding freshness [49].

Algae proteins, such as chlorella primordialis and spirulina, have also been tested in the manufacturing of plant-based meat alternatives to improve the product quality [50]. Microalgae proteins have good gelation, water absorption, fat absorption, emulsification and foaming abilities, all of which are beneficial in their application in tissue protein production. For example, the addition of spirulina protein in the production of tissue protein has resulted in products with good elasticity, flexibility and fiber properties [35]. Caporgno et al. [51] combined microalgae (30%) and soybean protein tissue to improve the nutritional value, appearance and color properties of the latter and obtain a good tissue protein with significant juiciness. Moreover, compared to other plant protein sources, microalgae have higher growth rates and can adapt to a wide range of growth conditions, making them an efficient source of material for protein production.

### 2.2. Effects of Lipids on the Structural Quality of Tissue Protein

During processing, lipids can impact the structural quality and sensory properties of textured proteins. Lipids can act as plasticizers or lubricants during the tissue protein molding process, which can reduce particle interactions in the material and lessen the friction between processing machine and material, resulting in improved texture, viscosity and integrity in the tissue protein product. The combination of different lipid concentrations and processing conditions can produce tissue proteins with differentiated structural characteristics to meet the specific needs of a variety of products [38].

The addition of lipids can also improve the organization of the legume tissue protein, resulting in a product with good fibrous structure and suitable hardness and flexibility. Jia et al. [52] found that the addition of 8% mixed plant oil could affect the protein interactions during wheat gluten protein extrusion by enhancing the polymerization of disulfide bonds, thereby improving the fibrous strength of the tissue protein and obtaining a well-organized product with improved continuity. However, it has been suggested that lipids can adversely affect the anisotropic fibrous structure of organized protein, and that excess lipid can inhibit the formation of fibrous structures, thus reducing the tensile strength and structural quality of tissue protein [53]. Kendler et al. investigated the effects of different lipid contents on the anisotropic fibril structure of tissue protein obtained by processing wheat gluten protein as a base material. No significant differences were observed between the 0% and 2% lipid content samples; however, the anisotropic fibril structure was reduced in the 4% lipid content samples [54]. This was due to the fact that as the lipid content increases, oil droplets aggregate and increase in size, interfering with the polymerization of gluten proteins and leading to a weakened protein network structure.

In general, the addition of lipids has varying effects on process kinetics. Further investigations are required to provide insight into the effects of lipids on processing conditions, protein polymerization, matrix rheological properties and their microstructure, as well as to clarify optimal quantities of lipid additions for various target products. The appropriate proportion of fats and oils can effectively improve fibrous strength and structural quality in the manufacture of organized products with different processing characteristics.

### 2.3. Effects of Moisture on the Structural Quality of Tissue Protein

Moisture affects both the fibrous structure and texture of tissue protein, contributing to the unfolding and arrangement of protein molecules during processing and promoting the formation of fibrous structures. It has previously been found that moisture content is a crucial factor affecting the formation of fibrous structure via the extrusion of lupin protein [55]. When moisture content is less than 40%, the hydration of lupin protein is incomplete, with ineffective cross-linking, and the resulting tissue protein fibers are poorly structured and prone to breaking [47]. In high moisture processing conditions, protein molecule aggregation relies mainly on hydrophobic interactions, while disulfide bonds replace hydrophobic interactions as the stabilizing force for protein molecule aggregation in low moisture-processing conditions [56]. During the increase in moisture content from 20% to 60%, disulfide bonding and hydrophobic interaction were found to occur synergistically, thereby facilitating increased fibrillation [57]. It has been suggested that the effect of moisture content on the hardness of tissue protein is most significant during extrusion [58]. Increased free moisture content has a lubricating effect on processing, promotes the fluidity of the material, lowers the processing intensity and pressure, reduces the expansion phenomenon, improves the compactness of the product structure, and results in a dense and juicy tissue protein structure [47]. Therefore, increasing the moisture content of the material is conducive to the promotion of protein-stretching denaturation, thereby improving the tissue protein quality of structural characteristics, including hardness, juiciness and other textural characteristics [59].

Moisture levels are also important for the production, storage and transportation of final products. In the most common extrusion process, for example, low moisture extrusion (with a moisture content of less than 40%) produces a relatively poor-tasting tissue protein, while high moisture extrusion (with a moisture content of more than 40%) retains moisture and produces a highly textured, elastic, tough, fresh and high-quality tissue protein with a texture similar to animal meat and an appearance and taste similar to cooked meat [60]. Furthermore, based on the difference in the moisture content of tissue protein, low-moisture tissue protein is less likely to spoil after drying and can be stored for a long time with a longer shelf life. However, the production cost of high-moisture tissue protein is high, and it has a short shelf life, with strict requirements for storage (during which it must be refrigerated) and transportation. Thus, further improvement is required.

### 2.4. Effects of Molding Process on the Structural Quality of Tissue Protein

In recent years, extrusion technology is still the main way to process tissue protein because it can form a fibrous structure similar to meat, and the research is relatively mature, in which high moisture extrusion has a better taste and gradually becomes the better choice. 3D printing technology is gaining the attention of scholars studying plant-based meat alternatives because of its personalization and nutritional customization and its ability to create products with complex structures or geometric shapes. Many scholars are gradually researching it in depth, expecting it to become an industrial production technology for plant-based meat alternatives [38,61,62]. Shear cell technology, spinning technology and refrigeration structure technology are also plant-based tissue-protein-processing technologies that distinguish themselves from the previous processing methods. This section of the review will focus on the technical processing principles and processes of five molding methods (with a key focus on extrusion). In addition, the effects of different technologies on the structural quality of tissue protein are analyzed, and the characteristics of the finished products of different molding methods are presented (Figure 2).

#### 2.4.1. Extrusion Technology

Extrusion technology integrates the mixing, homogenizing, aging and forming of materials through thermomechanical treatment and shear flow to form a fibrous structure that is similar to meat. It is dependent on the moisture content of the material, and therefore, when the moisture content of the material is lower than 40%, it is a low moisture extrusion process, and if it is higher than 40%, then it is a high moisture extrusion process [60]. Figure 2b is a graphical representation of the extrusion of final products with different moisture contents. The low moisture extrusion method does not require high protein content in raw materials [28]. Wet materials flow through the extruder at high temperature, the moisture in the material instantly turns to hot steam, the extrudate expands, and the structure becomes puffed sponge that must be rehydrated before use [65,66]. Low moisture extrusion is a more commonly used technology due to its wide applicability, high technical maturity and low equipment cost. High-moisture extrusion is a technique developed on the basis of low-moisture extrusion technology, which can better simulate the fiber structure of similar meat [67]. In the high-moisture extrusion process, the quality of the organized plant protein can be improved by increasing the protein concentration, and the high moisture content can ensure that the tissue protein will not break easily at high fiber strength, so the protein content of the raw material is usually more than 60%, and the tissue protein obtained via this process has higher moisture retention as well as high elasticity and toughness [68]. It is similar to meat products, with little loss of nutrients, and does not require rehydration before it can be eaten [69]. Due to its low energy consumption and excellent product performance, it is currently the most common method used in the production of juicy tissue protein in recombinant plant-based meat alternatives. While this technology and its market are still developing, the application potential of this approach is generally recognized by the industry as offering the most commercialization prospects [70].

#### 2.4.2. 3D Printing Technology

3D printing technology, also called additive manufacturing technology for rapid prototyping, refers to the manufacturing of solid 3D structures created from virtual models through computer-aided design [71]. In this type of tissue protein molding technology, molten raw materials are squeezed out from a movable nozzle, solidified and then stacked, layer-by-layer, to obtain prints that resemble the structure of meat [72]. This technique requires the modification of matrix via different cross-linking methods and pre-treatment to obtain a plant-protein-based printing material that can be squeezed easily from the nozzle with sufficient mechanical strength [73,74]. Shahbazi et al. used 3D printing technology to produce plant-based meat alternatives based on soybean isolate protein and explored the effect of introducing biosurfactants to achieve partial fat replacement and improved structural properties of 3D printed products [75]. A team of researchers intend to improve the texture of alternative meat by directly inserting hydrocolloid-based fibers into the protein matrix using a coaxial nozzle-assisted 3D food printer [76]. Although the 3D printing of plant-based meat alternatives has been explored to some extent, it is not yet fully developed and requires further progress to achieve the immediate and rapid fabrication of structurally superior plant-based meat alternatives.

#### 2.4.3. Other Technologies

Shear cell processing technology uses shear flow to mix raw plant protein materials and water. The processing time is controlled by adjusting the cylinder speed and temperature of the equipment. With anisotropic tissue proteins and a simple combination of shear force and heating, plant protein can be processed into homogenous, layered and multilayered fibrous structures [77]. Shear cell processing technology features a constant shear force and reduced mechanical energy dissipation, and is highly flexible in terms of required equipment. Moreover, it has great potential for growth in the field of tissue protein synthesis. The processed tissue protein has strong continuity in the shear direction, a long and slender fiber structure with a tight and dense interfiber, and is difficult to break [78].

Electrostatic spinning technology refers to the process in which nano-level oriented fibers are formed from plant protein polymer solutions in a nonwoven state under the breakdown of a high-voltage electrostatic field. However, the technological difficulty of this process is relatively high because proteins have a complex advanced structure, making them more difficult to destroy during spinning, and the pre-treatment requirements for raw materials are stringent [79]. These technical constraints limit the utilization of this method, as only some proteins are suitable for spinning technology, and, hence, it has relatively limited applications in tissue protein production.

Refrigeration structure technology produces porous fibrillar structures by, first, freezing protein emulsions and then removing the ice crystals to obtain tissue proteins that resemble the structure of animal muscle fibers and consist of many parallel and highly connected lamellar proteins [80]. The composite effect of the structural and physicochemical properties of different plant proteins results in differences in the structure of tissue proteins obtained from different plant-based composites. This technology has the capacity to form plant-based meat alternatives with unique textural contours and at a relatively low cost, making it suitable for small-scale industrial production [64].

## 3. Factors Affecting the Structural Quality of Simulated Fat

Simulated fat is an oil-containing protein-polysaccharide composite that mimics the structure and chewiness of animal fat, but it has a significantly lower lipid content. It can provide a filler structure and improve the texture and taste of recombinant plant-based meat alternatives. There are two main types of simulated fat, one that simulates entire fat with a distinct structure, and another that simulates the dispersed fat between tissue proteins and acts as an adhesive [81]. The dispersion of simulated fat can improve the textural, water, and oil-holding properties of recombinant plant-based meat alternative products so that their simulated fiber structure is juicier and has a delicate taste. Lipids and polysaccharides are the main factors influencing the structural quality of simulated fat because they can impart juiciness to the reconstituted plant flesh and fully exploit the binding effect. In addition, transglutaminase (TG) can promote cross-linking and has an important role in the formation of block simulated fat.

### 3.1. Effects of Lipids on the Structural Quality of Simulated Fat

Plant oils are suitable raw materials with which to simulate animal fats, as they are a relevant flavor source, providing proportional fatty acid composition for health benefits [82] as well as good processing characteristics to maximize the structural quality and sensory characteristics of products [83,84]. The quality of simulated fat is greatly influenced by the choice of lipid source materials, and therefore, the exploration of richer sources of lipids, appropriate proportions, specific applications and mechanisms related to the formation and improvement of structure by lipids are important directions for the progressive development of simulated fat.

The right proportion of lipids can effectively increase moisture retention in simulated fat, thereby improving the tenderness, texture and juiciness of recombinant plant-based meat alternatives [85]. In one recent study, 3.5% sunflower oil and konjac glucomannan were used as raw materials to develop a plant-based fishball analogue, the results of which showed that the addition of sunflower oil is beneficial to the texture and water retention of plant-based fishballs [86]. Similarly, Wi et al. [87] reported that the water retention of plant-based meat alternatives with added oil was superior to that of plant-based meat alternatives without the oil due to the enhanced interaction between the hydrocarbon side chains of the oil and the hydrophobic amino acids of the protein, which enhanced elastic behavior and gel strength and resulted in a greater water retention capacity. Overall, the substitution of animal fats with plant oils in recombinant plant-based meat alternatives has excellent health and processing advantages.

Different kinds of plant oils also possess different processing characteristics. The selection of a suitable oil based on the target product can improve the applicability and overall quality of simulated fat in recombinant plant-based meat alternatives. These different effects include those on the texture of simulated fat. In one study, as the concentration of palm oil increased, the hardness and viscosity of simulated fat gradually increased and the structure became more compact, whereas, with the increase in the concentration of soybean oil, simulated fat became softer [88]. The ratio of liquid fats to solid fats can be varied according to the needs of different products to improve the texture and flavor of simulated fat. Compared to the use of a single liquid plant oil, fat formulation has been shown to more effectively solve the texture and juiciness concerns of simulated fat. According to Dreher et al. [89] raw material contains 70% lipids (a mixture of rapeseed oil and solid hydrogenated rapeseed oil), wherein the rapeseed oil and its hydrides were incorporated into the plant protein network through emulsification to fully simulate animal adipose tissue, resulted in well-textured, juicy simulated fat, thereby laying the foundation for the development of fat systems for recombinant plant-based meat alternatives.

### 3.2. Effects of Polysaccharides on the Structural Quality of Simulated Fat

Polysaccharides are an important raw material in the processing of simulated fat. A protein-polysaccharide gel network structure can be obtained via the compounding with protein, providing good water and oil-holding properties and resulting in simulated fat with juicy and lubricious textures. In this section of the review, the characteristics and optimization of different polysaccharides in previous studies are analyzed from the viewpoint of the influence of polysaccharides on the structural quality of block simulated fat with adhesive effect, providing a foundation for the selection of polysaccharide-based raw materials for simulated fat. Starch is a polysaccharide raw material that commonly acts as a filler, facilitating the combination of protein, water and lipid components in the product, maintaining structural stability, and generating a mimicked fat with good formability by combining water and lipids, mostly through physical means [38]. It has been reported that when simulated fats were prepared from coconut and soybean oils, the addition of appropriate amounts of potato starch increased the dynamic modulus and hardness, reduced the meltability of the simulated fat, and achieved similar levels of taste and sensory qualities of commercial beef and pork fats [90]. Bon-Yeob and Gi-Hyung [91] demonstrated that the addition of 10% corn starch improved cohesion between protein molecules, resulting in a simulated fat with higher water absorption, elasticity and adhesion. This simulated fat, with its adhesive effect, can fill in the tissue protein interstices and fully imitate the fat dispersal in lean meat products to obtain recombinant plant-based meat alternatives with tight structure and good juiciness.

Methylcellulose and konjac glucomannan are also important raw materials in the production and processing of simulated fat. Methylcellulose absorbs water, swells during processing, and has good water-holding and gelation properties. After heating, the viscosity increases significantly, which is conducive to improving the texture and juiciness of simulated fat, which makes it a popular binder for the formulation of simulated fat in plant-based hamburger [92]. Huang et al. [93] found that the addition of 4% konjac glucomannan to solid block simulated fat with soybean protein and coconut oil as the main ingredients resulted in a similar appearance to pork fats, with more desirable functional properties in terms of mechanical properties and oral tribology, and better chewiness and mouthfeel scores.

### 3.3. Effects of TG on the Structural Quality of Simulated Fat

Transglutaminase (TG) is a commonly used synergist and cross-linking agent in simulated fat processing, in which it can play a bonding and cross-linking role, thereby promoting molding and improving the resulting simulated whole animal fat [94]. TG acts on the acyl transfer reaction between the γ-amino group of the glutamine (Gln) residue and the ε-amino group of the lysine (Lys) residue [95]. Protein can be modified by adding amine groups to promote intermolecular cross-linking under its induced covalent cross-linking effect, thus altering its molecular structure and resulting in improved simulated fat gels, textures and other functional properties [96,97]. A 2021 study have found that TG can provide adhesion for the TG-induced gel, with covalent (isopeptide) bonds used as an adhesive to link tissue protein and simulated fat, which led to sufficient cohesion to not fragment in the assembled bacon analogues. [98]. Wen et al. found that TG could promote protein aggregation, increase the stiffness, elasticity and cohesiveness of gels, contribute to the formation of a more homogeneous and denser three-dimensional network, and notably enhance the resistance of TG-induced bitter almond protein gels to simulated gastric juice [99]. Dreher et al. [100] manipulated plant oils and their hydrides into the network structure of plant proteins via emulsification, and then induced covalent cross-linking between proteins by TG, obtaining whole simulated fat with a certain hardness crushed to pellets with good granularity and used in the preparation of plant-based salami.

## 4. Factors Affecting the Overall Structural Quality of Recombinant Plant-Based Meat Alternatives

Tissue proteins can be assembled and combined with simulated fat and binders to obtain quality characteristics that match the product attributes of particular meat products. This is made possible by combining the analysis of animal meat composition with an understanding of the histochemical ability of plant proteins [19]. Tissue protein and simulated fat, as the major components of recombinant plant-based meat alternatives, play critical roles in its structural quality and can be blended in a variety of ways depending on the simulated product [10]. Moreover, the structural quality and simulation effect of recombinant plant-based meat alternatives are highly dependent on the choice of assembly and molding processes.

### 4.1. Effects of Tissue Protein on the Structural Quality of Recombinant Plant-Based Meat Alternatives

Tissue protein in plant-based meat alternatives provides structure and chewiness similar to those of lean meat fiber. It is also the skeleton structure for building recombinant plant-based meat alternative products and has a decisive role in stimulating the structural characteristics of animal meat products [18]. Suitable raw materials can be chosen on the basis of various tissue protein characteristics in order to properly imitate the structural characteristics of various animal meat products and generate recombinant plant-based meat alternatives with excellent quality.

Under the same processing conditions, the soybean tissue protein after silk tearing is thicker, with greater toughness and hardness, suitable for the production of chewable recombinant plant-based meat alternatives with a stronger fiber structure [69]. Wheat-based protein is finer, longer and more brittle after dismantling, making it ideal for the production of recombinant plant-based meat alternatives with a delicate flesh and strong continuity of fiber structure [66]. Pea-protein-based products are less filamentous, less elastic and have a pea bean flavor after shredding, making them suitable for the production of recombinant plant-based meat alternatives suitable for higher processing precision, lower requirements for fiber structure, but higher flavor requirements [101].

### 4.2. Effects of Simulated Fat on the Structural Quality of Recombinant Plant-Based Meat Alternatives

The simulated fat helps to refine the structure of the recombinant plant-based meat alternatives to adequately mimic the taste of animal meat products. It has been demonstrated that by optimizing the formulation, block simulated fat with sensory properties similar to pork fat, such as hardness, flexibility and chewiness, can be created for the assembly of reconstituted plant-based meat alternatives with improved sensory qualities [93]. Simulated fat has favorable water and oil-holding properties, and may, thus, provide recombinant plant-based meat alternatives with a delicate and juicy feel, maximizing the bonding effect, and facilitating assembly and molding. It has been shown that the pulverization of block simulated fat can be used to bind tissue protein particles for the preparation of plant-based salami [100]. Products produced via this method are assembled and shaped by the binding action and have a meat-like granularity, juiciness and a significant simulated effect.

Simulated fat based on plant oil enables recombinant plant-based meat alternatives to maintain similar organoleptic properties to those found in animal fat, reducing the negative impact of the process on organoleptic properties and improving product structure and quality stability. Nieto et al. [102] prepared simulated fat from olive oil to replace some animal fats in sausages and monitored the emulsion stability, color and lipid oxidation of the sausages, finding that the deterioration of sausage quality was delayed, shelf life was extended and quality stability was significantly improved. Shahbazi et al. [8] used soybean isolate protein, rapeseed oil and modified inulin as raw materials to prepare simulated fat for the construction of 3D-printed low-fat plant-based meat alternatives, thereby greatly improving the emulsion stability while reducing the lipid content, and the resultant printed plant-based meat alternatives was shown to exhibit good continuity and formability.

## 5. Future Prospects

The recombinant plant-based meat alternatives market is currently expanding worldwide, with the number of commercially available products growing steadily. Likewise, the manufacturing processes are becoming more industrialized and advanced; however, in comparison to animal meat, recombinant plant-based meat alternatives still have insufficient flavor, a large structural gap, and poor sensory attributes and chewing sensations. Its structural quality is one of the important factors affecting consumers’ preferential purchasing decisions. Furthermore, there are relatively few whole reconstituted plant-based meat alternative products available that are suitable for a variety of cooking methods, and the sector is largely undeveloped, with significant research gaps and market potential.

(1) The future development of recombinant plant-based meat alternatives should be devoted to optimizing the structural quality and sensory characteristics of tissue protein. There is a need to expand the sources of plant proteins, explore in depth the new characteristic proteins and improve the quality of tissue proteins via the use of different processing of characteristic proteins for compounding, through enzyme modification and other methods. Moreover, the quality of tissue proteins could be improved by adding proteins with better gel ability to assist the main protein raw materials.

(2) For the optimal molding of tissue protein, it is necessary to improve the inclusiveness of production equipment for raw materials and the level of processing technology, and to reduce production costs, as well to explore the simulation of meat fiber structure and chewing sensation in depth. Furthermore, the application of high moisture extrusion and 3D printing technologies with excellent production quality and good development prospects in the production of tissue protein should be promoted.

(3) The nutritional composition and processing characteristics of different varieties of lipids and polysaccharides warrant in-depth investigation, and the optimal additions and related processes should be clarified to optimize their use in simulating fats and binders. It is also important to explore more abundant sources of lipids and polysaccharides, and to optimize the nutritional ratios, functional properties and specific applications of fat substitutes, as an important direction for the future development of plant-based meat alternatives. In addition, the effect of TG on simulated fat structures and related mechanisms should be researched to promote the application of TG to improve structural quality and enhance simulation effectiveness.

(4) According to the plant-based meat product, different types of simulated fat or binders should be selected to optimize structural quality and to obtain high-quality plant-based meat alternatives with good juiciness and chewiness. The development of simulated fat binders is under consideration for our own future study to obtain a protein–fat interphase in recombinant plant-based meat alternatives with good texture, while developing the hierarchy of a complete meat structure.

(5) Whole recombinant plant-based meat alternatives offer a wide range of culinary applicability and scenarios, and they should be developed in-depth. In the future, the simulation accuracy of tissue protein, fat and the skin-layering structure of whole meat should be improved to ensure that it can undergo corresponding changes in cooking while retaining its distinct taste and texture. Recombinant plant-based meat alternatives are an important future research direction in plant meat technology that deserves further investigation.

## Figures and Tables

**Figure 1 foods-11-02202-f001:**
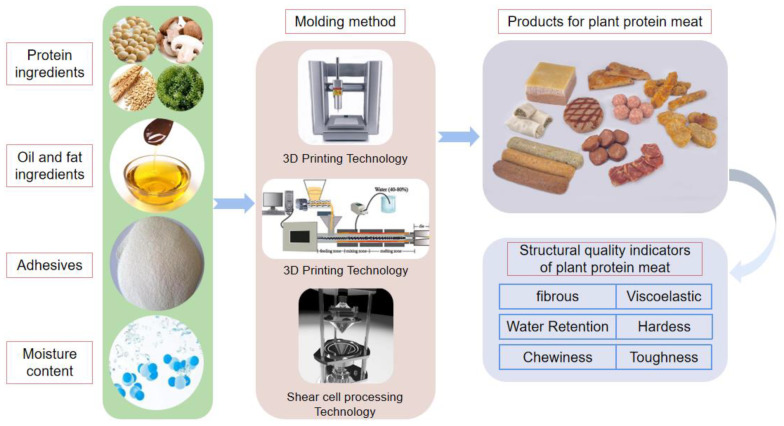
Factors influencing the structural quality of recombinant plant-based meat alternative [15,16,17].

**Figure 2 foods-11-02202-f002:**
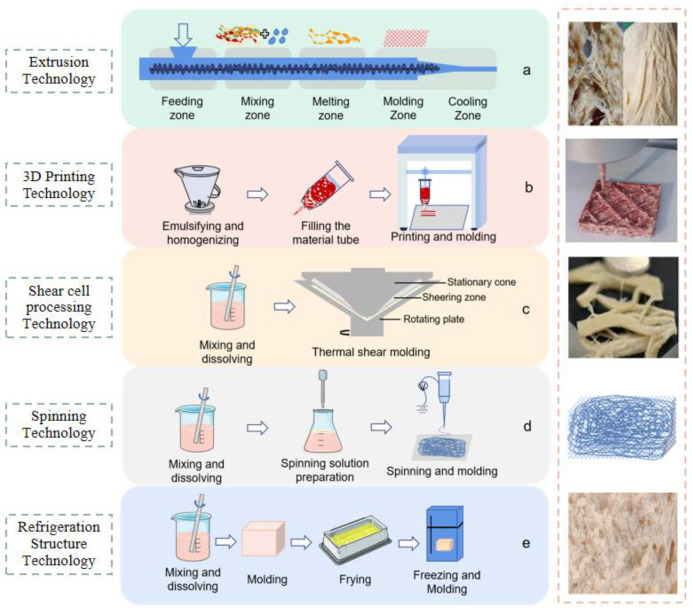
Schematic diagram of the processing principle, process and finished products of different molding technologies [16,63,64]. (**a**) Schematic diagram of the processing principle, process and finished products of extrusion technology; (**b**) Schematic diagram of the processing principle, process and finished products of 3D printing technology; (**c**) Schematic diagram of the processing principle, process and finished products of shear cell processing technology; (**d**) Schematic diagram of the processing principle, process and finished products of electrostatic spinning technology; (**e**) Schematic diagram of the processing principle, process and finished products of refrigeration structure technology).

**Table 1 foods-11-02202-t001:** Comparison of the properties of different protein raw materials.

Protein sources		Characteristics and Applications	
Legume Protein	Soybean Protein	Nutritional Characteristics	High protein content (approx. 40%); rich in amino acids; balanced ratio, close to WHO/FAO recommended standards [21]
		Processing Characteristics	Excellent processing properties, including good gelation, emulsification, water retention and anisotropic fiber structure [22]
		Limiting Factors	Soybean odor (limits consumer’s acceptance); allergenicity (allergenic to people with protein allergies, especially infants and children [23])
		Specific Applications	Impossible Food (USA) launched a plant-based burger patty made from soy protein concentrate; Qishan Food (China) introduced a plant-based steak with soybean isolate protein as the main ingredient
	Pea Protein	Nutritional Characteristics	High carbohydrate content; 30% protein; amino acid ratio close to WHO/FAO recommended standards; high lysine content; lack of methionine; high bioavailability [24]
		Processing Characteristics	Pea protein can absorb more fat with the same water content and has good water- and oil-holding properties [25]
		Limiting Factors	Relatively weak heat-induced gelling ability (need to optimize the gel strength of the product by changing the stability of interprotein molecular force interaction through salt ions or processing conditions [26])
		Specific Applications	Beyond Meat (USA) launched a plant-based meat burger with pea protein as the main ingredient; Shuangta Foods (China) launched a plant-based meat chicken nugget with pea tissue protein and pea isolate protein as ingredients
Grain Protein	Wheat Protein	Nutritional Characteristics	Low protein content (approx. 13%); unsatisfactory amino acid composition; lack of lysine; low digestibility [27]
		Processing Characteristics	Excellent water absorption; viscoelasticity; film-forming; thermosetting; and other processing characteristics [28]
		Limiting Factors	Allergenicity (intake of wheat bran protein by intolerant individuals can trigger reactions such as celiac disease, nonceliac gluten allergy and wheat allergy [29])
		Specific Applications	Field Roast (USA) launched a plant meat burger made from wheat isolate protein; wheat protein has been added to plant meat products such as Jin Zi Ham plant meat patties (China), Impossible Burger (USA), and Garden Meatless Meat Balls (Canada)
Non-Plant Classified Protein	Fungal Protein	Nutritional Characteristics	Protein content of approx. 15%; amino acid ratio close to WHO/FAO recommended standards; high sulfur content; glutamic acid for fresh taste [30,31]
		Processing Characteristics	The texture and chewiness of plant-based meat alternatives developed from fibrous fungal protein in the form of mycelium is similar to that of meat products [32]
		Limiting Factors	The core technology development of bacteriophage protein meat processing is not yet perfect, and standardization and industrialization have not yet been achieved
		Specific Applications	Quorn Food (UK) introduced plant-based fried chicken nuggets made from fungal protein
	Algal Protein	Nutritional Characteristics	Rich protein content (50–71%); balanced amino acid composition (the essential amino acid ratio is comparable to casein in milk); high bioavailability; comparable to soy protein and better than wheat gluten protein [33]
		Processing Characteristics	Excellent abilities to gel, absorb water and fat, emulsify and foam [34]
		Limiting Factors	Algae odor (earthy and moldy algae-like odor [35]); biotoxicity (potential biotoxicity and risk of toxin accumulation due to water contamination [36])
		Specific Applications	Sophie’s Bionutrients (Singapore) launched burger patties made from microalgae protein

## Data Availability

The data presented in this study are available on request from the corresponding author.

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
