# Peer review of "Ingredients and Process Affect the Structural Quality of Recombinant Plant-Based Meat Alternatives and Their Components"

_foods, 2022, doi:10.3390/foods11152202_

Round 1

Reviewer 1 Report

The submitted manuscript is review covering recent 5 years in advances in plant based meat alternatives. The aim of the review is clear and the structure as well. The introduction is simple and lead well to the topic. The rest of the manuscript is easy to read but is to general. The reader gets scarce information about most common protein sources and processing techniques for plant based meat analogues. The focus on recent advances within last 5 years is barely visible. Summarizing the manuscript is a well prepared paper but rather suitable for a book chapter rather than review in scientific journal.

Detailed remarks:

Line 49 style - both alternative and products are in plural

Line 63 as above

Line 57-85 no source two paragraphs of text i

Line 68 oriental?

Figure 1 molding techniques should be also named

Table 1 is barely readable, please reedit it

Line 131 compared to soybean peas are low in protein but they are abundant source of protein (please rephrase)

Line 324-I think this topic could be elaborated there are various interesting articles regarding 3d printing in foods as well as other MDPI journals

Line 425 please rephrase it “sounds” as konjac glucomannan is a cellulose derivative

Line 443-446 not relevant for plant based meat

Line 456-473 no source for 2 paragraphs

Line 485-487 repetition of information

Reviewer 2 Report

This review reported that the ingredients and process affect the structural quality of recombinant plant-based meat alternative and its components. The review is interesting and easy to read. Some specific comments are as follows:

1. The description of the abstract should be more concise.

2. L307-309 Rewrite this sentence. Why protein content of the raw material must be greater than 60%?

3. L328 Thermal shear processing should be shear cell processing.

4. L355: lipid-protein-polysaccharide complex? This description is inaccurate.

Round 2

Reviewer 1 Report

The manuscript has been improved. However, one major issue was omitted during initial review – the methodology of the review. Please state keywords and databases used for the literature search and if possible what were the inclusion or exclusion factors for manuscript to be considered.
